# Learning From Multi-Expert Demonstrations: A Multi-Objective Inverse Reinforcement Learning Approach

## Abstract

Imitation learning (IL) from a single expert's demonstration has reached expert-level performance in many Mujoco environments. However, real-world environments often involve demonstrations from multiple experts, resulting in diverse policies due to varying preferences among demonstrators. We propose a multi-objective inverse reinforcement learning (MOIRL) approach that utilizes demonstrations from multiple experts. This approach shows transferability to different preferences due to the assumption of a common reward among demonstrators. We conducts experimental testing in a discrete environment Deep Sea Treasure (DST) and achieved a promising preliminary result. Unlike IRL algorithms, we demonstrate that this approach is competitive across various preferences in both continuous DST and Mujoco environments, using merely a single model within the SAC framework instead of $n$ models for each distinct preference.

## 1 Introduction

Multi-objective inverse reinforcement learning (MOIRL) is crucial in the field of robot control. In certain real-world scenarios, demonstrations are gathered from various experts due to the lack of data. For example, the intelligent control systems for military drones or robotic arms stepping in for doctors to perform rare surgeries. In such contexts, demonstrations are not only scarce but also hard to obtain and therefore involving multiple experts. It's inevitable to see two or more individuals can have totally distinct preferences while engaging in the same task. Agents operating military drones may need to strike a balance between aggressiveness and the risk of being destroyed, whereas doctors performing surgeries may consider both precision and time efficiency.

Learning from multi-expert demonstrations can be essentially achieved by repeatedly running inverse reinforcement learning (IRL) multiple times. However, this approach can be inefficient and may compromise performance because of the scarcity of demonstrations from a single expert. Most importantly, it lacks cooperation among experts. As previously mentioned, while in the same task, as the main motivation of this work, we believe the concept of shared knowledge instead of running multiple independent IRL algorithms will improve the results. Within the framework of MOIRL, we assume there's one common vectorized reward among experts. Preference can influence the scalar reward, which means different policies only come from different preferences.

Traditional IRL typically learns policy by first learning a reward function, introducing a challenging max-min optimization problem. In contrast, MOIRL can benefit from shared knowledge, which is the common vectorized reward in our case. It can be considered as an additional constraint in optimization problem. In discrete case, we repeatedly solve the common reward with consensus alternating direction method of multipliers (ADMM), incorporating both demonstrations and the current policies of agents to iteratively refine policies and reward through RL and IRL. To enforce the common reward constraint in continuous environments, we import the settings of multi-objective into the framework of IQ-Learn, transforming the reward consensus constraint into the objective, acting as a penalty term. Our proposed MOIRL framework brings several advantages. It can enhance the availability of collected data. Furthermore, by the means of common reward, our model has ability to generalize to other preferences absent from demonstrations.

We summarizes our contributions as follows:

- We utilize consensus ADMM to satisfy the common reward constraint, resulting in a promising experimental foundation on discrete deep sea treasure (DST) environment.
- We extend the IQ-learn framework to the field of MOIRL, building connections among heterogeneous agents during training, thereby allowing a more flexible policy for collecting demonstrations from various experts.
- We show the transferability of our model by twisting the SAC networks with additional preference input.

## 2 RELATED WORK

**IL/IRL with single-expert:** It's obvious that naively solving a max-min optimization problem through nested loops of RL and IRL is impractical as it costs a lot of computational resources. As the very first work taking a great step, Ho & Ermon (2016) proposed a more general and practical framework base on the insight that IRL is essentially a dual of an occupancy measure matching problem, which learns a policy as the generator trying to fool the discriminator, drawing an analogy with generative adversarial networks (Goodfellow et al., 2014). However, the adversarial learning can still be inefficient. Recently, Garg et al. (2022) has proposed a Q-learning approach that get away with adversarial optimization process. They utilized the energy-based policy and inverse soft Bellman operator to replace the original objective into a single maximization problem over Q space. This approach learns policy and retrieves reward function in a direct manner.

**IL/IRL with multi-expert:** In recent years, more and more works start focusing on IL and IRL with multi-expert demonstrations due to several reasons. As an extension of GAIL, Li et al. (2017), and Hausman et al. (2017) introduce a latent variable to disentangle trajectories that may arise from a mixture of experts. However, these approaches are constrained by the limitations of IL, such as the inability to adapt to environmental changes and excessive reliance on the quantity and quality of experts. As traditional IRL treats demonstrations homogeneously, Beliaev et al. (2022) has taken the expertise of demonstrators into account. They estimate the expertise of demonstrators and learn the optimal policy by fitting policies of demonstrators with negative log-likelihood loss. There are handful related works that also take multi-objective into consideration. Kishikawa & Arai (2021) introduced Non-Negative Matrix Factorization (Lee & Seung, 2000) into MOIRL by treating common reward vector as the basis matrix to solve the common reward vector and weights together. The method is still an indirect and restricted approach as it needs to run single-objective IRL first and is only applicable on discrete environment. Kishikawa & Arai (2022) has further proposed a framework to estimate the common reward vector and weight via neural networks base on the reward-seeker principle. Furthermore, Chen et al. (2020) utilized network distillation to distill common knowledge from individual strategy preferences to the task reward. However, MSRD requires training in an all-at-once manner and lacks the capability to accommodate lifelong learning. In response to this limitation, Chen et al. (2022) adeptly models new demonstrations by treating them as combinations of previously acquired prototypes. This solves the challenge of effectively representing a large number of demonstrations. However, the biggest problem of their works is it completely ignores the computational cost because it still needs to run IRL $n$ times. In contrast, Our work adopts a single model architecture. The core idea of our work is sharing knowledge within a single model. The reason behind this is that these heterogeneous experts are engaging in the same task, with their only differences lying at the preference level.

## 3 PRELIMINARIES

**Notations**  In this paper, $\Pi$, $\mathcal{R}$ represent the policy space and reward space, we use $\pi_{E_i}$ and $\pi_i$ to denote the policy of $i$th expert and the learned policy respectively. For a policy $\pi \in \Pi$, occupancy measure $\rho_\pi : \mathcal{S} \times \mathcal{A} \to \mathbb{R}$ is defined as $\rho_\pi = (1 - \gamma)\pi(a|s)\sum_{t=0}^{\infty} \gamma^t P(s_t = s|\pi)$. For brevity, we refer to $\rho_{\pi_i}$ as $\rho_i$.

**Multi-Objective Markov Decision Process (MOMDP).**  We consider the environment formulated by the tuple $(\mathcal{S}, \mathcal{A}, p_0, \mathcal{P}, \boldsymbol{r}, \gamma)$, where $\mathcal{S}$, $\mathcal{A}$ denote state and action spaces. $p_0$ is the distribution of initial state $s_0$, $\mathcal{P} : \mathcal{S} \times \mathcal{A} \times \mathcal{S} \to [0, 1]$ is the transition function of the environment, $\boldsymbol{r} : \mathcal{S} \times \mathcal{A} \to \mathbb{R}^d$ is reward function in vector form where $d$ represents the number of objectives, $\gamma \in (0, 1)$ is the discount factor.

The vectorized reward can be scalarized by a scalarization function $f_\omega : \mathbb{R}^d \to \mathbb{R}$ (Abels et al., 2019). In this paper, we focus on the linear scalarization function, that is ,

$$f_\omega(\boldsymbol{r}(s,a)) = \omega^T \cdot \boldsymbol{r}(s,a) = r_s(s,a) \tag{1}$$

where $r_s$ is the scalarized reward function, $\omega$ is a vector with $d$ non-negative entries that adds up to 1, representing the preference of the expert.

**Alternating Direction Method of Multipliers (ADMM).**  ADMM is an iterative algorithm used to solve distributed optimization problems. Its fundamental concept involves transforming the original optimization problem into multiple decomposed sub-problem. By alternately updating these sub-problem, ADMM approaches the optimal solution and eventually achieves a global solution.

The ADMM method can address global variable consensus optimization problem through distributed optimization. Consider the scenario where there is a single global variable, and the objective and constraint terms are divided into N parts: minimize $\sum_{i=1}^{N} f_i(x)$. This problem can be reformulated by introducing local variables $x_i$ and a shared global variable z as follows:

$$\begin{aligned} \text{minimize} \quad & \sum_{i=1}^{N} f_i(x_i) \\ \text{subject to} \quad & x_i - z = 0, \ i = 1, \dots, n. \end{aligned} \tag{2}$$

Each iteration of ADMM can be simplified to the following updates:

$$\begin{aligned} x_i^{k+1} &:= \underset{x_i}{\arg\min}(f_i(x_i) + (\rho/2)||x_i - \bar{x}^k + u_i^k||_2^2) \\ u_i^{k+1} &:= u_i^k + x_i^{k+1} - \bar{x}^{k+1}. \end{aligned} \tag{3}$$

where $\bar{x}^k = (1/N)\sum_{i=1}^{N} x_i^k$. It's evident that the updates of $x$ and $u$ can both be implemented using distributed computing.

**Inverse Reinforcement Learning (IRL).**  The goal of IRL is to find the reward function maximizing the difference between expected cumulative rewards under occupancy measures of expert and agent in the outer loop while seeking for a policy that minimizes negative expected cumulative reward of the agent in the inner loop.

$$\max_{r \in \mathcal{R}} \min_{\pi \in \Pi} \mathbb{E}_{\rho_E}[r(s,a)] - \mathbb{E}_\rho[r(s,a)] \tag{4}$$

While it can easily have multiple optimal policies satisfying the formulation for a given reward function, maximum-entropy IRL (Ziebart et al., 2008) is proposed to tackle down the ambiguity, along with a reward regularizer $\psi$ to prevent overfitting:

$$\max_{r \in \mathcal{R}} \min_{\pi \in \Pi} \mathbb{E}_{\rho_E}[r(s,a)] - \mathbb{E}_\rho[r(s,a)] - H(\pi) - \psi(r) \tag{5}$$

**Inverse soft Bellman operator.**  Garg et al. (2022) proposed inverse soft Bellman operator $\mathcal{T}^\pi$ to further characterize the relation between reward and $Q$ space. It's defined as:

$$(\mathcal{T}^\pi Q)(s,a) = Q(s,a) - \gamma \mathbb{E}_{s' \sim \mathcal{P}(\cdot|s,a)}[V^\pi(s')]$$

where $V^\pi(s) = \mathbb{E}_{a \sim \pi(\cdot|s)}[Q(s,a) - \log \pi(a|s)]$ is soft value function. $r$ and $Q$ have one-to-one correspondence under the definition of $\mathcal{T}^\pi$.

By leveraging inverse soft Bellman operator and an appropriate definition of reward regularizer $\psi$, equation 5 can be further simplified as (Garg et al., 2022):

$$\mathcal{J}(\pi, Q) = \mathbb{E}_{\rho_E}[\phi(Q(s,a) - \gamma \mathbb{E}_{s' \sim \mathcal{P}(\cdot|s,a)} V^\pi(s'))] - \underbrace{(1 - \gamma)\mathbb{E}_{p_0}[V^\pi(s_0)]}_{V_0 \text{ loss}} \tag{6}$$

where $\phi$ is a concave function and $p_0$ is the initial state distribution. The second term can be further replaced by $\mathbb{E}_{(s,a) \sim \mu}\left[V^\pi(s) - \gamma \mathbb{E}_{s' \sim \mathcal{P}(\cdot|s,a)} V^\pi(s')\right]$, where $\mu$ represents any valid occupancy measure.

# 4 METHOD

## 4.1 MOIRL WITH CONSENSUS ADMM (DISCRETE CASE)

In the subsequent section, we integrate the ADMM concept and the occupancy measure to reach the global reward function of MOIRL algorithm:

Initially, we extend equation 4 to accommodate $n$ experts with multi-objective reward:

$$\mathcal{J}(\pi_1, \boldsymbol{r}_1, \omega_1, ..., \pi_n, \boldsymbol{r}_n, \omega_n) = \max_{\boldsymbol{r}_1,...,\boldsymbol{r}_n \in \mathcal{R}} \min_{\pi_1,...,\pi_n \in \Pi} \sum_{i=1}^{n} \omega_i^T \left( \mathbb{E}_{\rho_{E_i}}[\boldsymbol{r}_i(s,a))] - \mathbb{E}_{\rho_i}[\boldsymbol{r}_i(s,a)] \right)$$

where $\omega_i$ is the preference of expert $i$. Note that reward functions here are optimized separately. With the goal of deriving a common reward function, we incorporate the consensus ADMM, treating reward function as consensus:

$$\mathcal{J}(\pi_1, \boldsymbol{r}_1, \omega_1, ..., \pi_n, \boldsymbol{r}_n, \omega_n) = \max_{\boldsymbol{r}_1,...,\boldsymbol{r}_n \in \mathcal{R}} \min_{\pi_1,...,\pi_n \in \Pi} \sum_{i=1}^{n} \omega_i^T \left( \mathbb{E}_{\rho_{E_i}}[\boldsymbol{r}_i(s,a)] - \mathbb{E}_{\rho_i}[\boldsymbol{r}_i(s,a)] \right)$$

$$\text{subject to} \quad \boldsymbol{r}_i = \boldsymbol{r}$$

Given initial $\pi_1^0, ..., \pi_n^0$, this reward consensus can be iteratively solved by:

$$\boldsymbol{r}_i^{k+1} = \arg\max_{\boldsymbol{r}_i} \omega_i^T \left( \mathbb{E}_{\rho_{E_i}}[\boldsymbol{r}_i(s,a)] - \mathbb{E}_{\rho_i}[\boldsymbol{r}_i(s,a)] \right) - (\rho/2)||\boldsymbol{r}_i - \bar{\boldsymbol{r}}^k + u_i^k||_2^2$$

$$u_i^{k+1} = u_i^k + \boldsymbol{r}_i^{k+1} - \bar{\boldsymbol{r}}^{k+1} \tag{7}$$

where $\bar{\boldsymbol{r}}^k = \frac{1}{n} \sum_{i=1}^{n} \boldsymbol{r}_i^k$. With the common reward solved, we train $n$ agents by running RL algorithm, looking for solving $\pi_1^j, ..., \pi_n^j$ accordingly. By repeating this procedure for enough $j$ rounds (Note that $j$ rounds here is different from $k$ iterations in ADMM), it's expected that this solved reward is getting closer and closer to the true reward.

### 4.1.1 LEARNING REWARD OF ABSORBING STATES

While this form of adversarial imitation learning may seem quite simple and intuitive, it can suffer from the issue of reward bias, which can significantly impact performance (Kostrikov et al., 2018). The problem lies in reward function, as it implicitly provides a survival bonus, leading to an non-ending loop in the agent's trajectory until it reaches maximum timesteps of the environment. The survival bonus encourages lasting longer in an episode, which is basically contradicting to the environments with step cost, or the environments with variable-length episodes. To address this, we simply learn a reward for absorbing states. Whenever the agent reaches a terminal state, it will transit to the corresponding absorbing state and stay until reaching maximum timesteps, ensuring a fixed-length episode.

### 4.1.2 EXPERIMENTAL TEST

We evaluate our algorithm on a simple task: Discrete Deep Sea Treasure(DST). In the case of a $6 \times 6$ mini-map, we conduct tests by learning from two experts with preferences [0.1, 0.9] and [0.9, 0.1] respectively. For the default $11 \times 12$ map, we learn from three experts with preferences [0.1, 0.9], [0.5, 0.5], and [0.9, 0.1].

As depicted in Figure 1, all agents reach near-optimal reward within 10 rounds in both configurations of the maps. This demonstrates the promising performance of our algorithm in the DST environment, indicating the idea of learning a common reward function among agents actually helps.

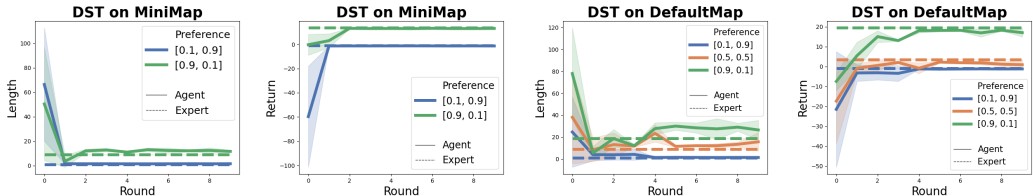

Figure 1: **Comparison of our algorithm and the optimal policy.** We present our results in terms of return and length, with averaging across 5 different seeds. A *Round* is defined as the completion of one iteration incorporating the MOIRL algorithm with consensus ADMM and the RL algorithm, specifically PPO.

### 4.2 MULTI-OBJECTIVE INVERSE SOFT-Q LEARNING (CONTINUOUS CASE)

#### 4.2.1 MULTI-EXPERT OBJECTIVE

By considering optimizing $n$ experts together with common reward constraint, we have our optimization problem to be (from equation 5):

$$\mathcal{J}(\pi_0, \boldsymbol{r}_0, \omega_0, ..., \pi_n, Q_n, \omega_n) = \max_{\boldsymbol{r}_0,...,\boldsymbol{r}_n \in \mathcal{R}} \min_{\pi_0,...,\pi_n \in \Pi} \sum_{i=0}^{n} \left[ \mathbb{E}_{\rho_{E_i}}[\omega_i^T \cdot \boldsymbol{r}_i(s,a)] - \mathbb{E}_{\rho_i}[\omega_i^T \cdot \boldsymbol{r}_i(s,a)] \right.$$
$$\left. - H(\pi_i) - \psi(\omega_i^T \cdot \boldsymbol{r}_i) \right] \quad \text{subject to} \quad \boldsymbol{r}_i = \boldsymbol{r}.$$

Because $\boldsymbol{r}_i$ involves both $\pi_i$ and $Q_i$ for every expert $i$, the analysis can become too complicated. We ease the difficulty by translating explicit constraint to implicit penalty term, which is l2 norm between the difference of each individual reward vector $\boldsymbol{r}_i$, we have:

$$\mathcal{J}(\pi_0, \boldsymbol{r}_0, \omega_0, ..., \pi_n, Q_n, \omega_n) = \max_{\boldsymbol{r}_0,...,\boldsymbol{r}_n \in \mathcal{R}} \min_{\pi_0,...,\pi_n \in \Pi} \sum_{i=0}^{n} \left[ \mathbb{E}_{\rho_{E_i}}[\omega_i^T \cdot \boldsymbol{r}_i(s,a)] - \mathbb{E}_{\rho_i}[\omega_i^T \cdot \boldsymbol{r}_i(s,a)] \right.$$
$$\left. - H(\pi_i) - \psi(\omega_i^T \cdot \boldsymbol{r}_i) \right] - \sum_{i=0}^{n-1} ||\boldsymbol{r}_i - \boldsymbol{r}_{i+1}||_2$$

It can be further split into $n$ separate optimization objectives, we can optimize agent $i$ with objective:

$$\mathcal{J}(\pi_i, \boldsymbol{r}_i, \omega_i) = \max_{\boldsymbol{r}_i \in \mathcal{R}} \min_{\pi_i \in \Pi} \overbrace{\mathbb{E}_{\rho_{E_i}}[\omega_i^T \cdot \boldsymbol{r}_i(s,a)] - \mathbb{E}_{\rho_i}[\omega_i^T \cdot \boldsymbol{r}_i(s,a)] - H(\pi_i) - \psi(\omega_i^T \cdot \boldsymbol{r}_i)}^{\text{Same as equation 5}}$$
$$- \beta \sum_{j=i-1}^{i} ||\boldsymbol{r}_j - \boldsymbol{r}_{j+1}||_2$$

where $\beta$ is the constraint coefficient controlling the importance of the common reward constraint.

By replacing $\omega_i^T \cdot \boldsymbol{r}_i(s,a)$ with scalar reward $r_s$ (from equation 1), it can be simplified as:

$$\mathcal{J}(\pi_i, Q_i, \omega_i) = \mathbb{E}_{\rho_{E_i}} \left[ \phi \left( \omega_i^T \cdot (Q_i(s,a) - \gamma \mathbb{E}_{s' \sim \mathcal{P}(\cdot|s,a)} V^{\pi_i}(s')) \right) \right]$$
$$- (1-\gamma) \mathbb{E}_{p_0}[\omega_i^T \cdot V^{\pi_i}(s_0)] - \beta \sum_{j=i-1}^{i} ||\boldsymbol{r}_j - \boldsymbol{r}_{j+1}||_2 \tag{8}$$

#### 4.2.2 UPDATE STRATEGY AND PRACTICAL ALGORITHM

**Critic network update:** We use $Q(s, a, \omega_i) \approx Q_i(s, a)$, which allows us to learn and estimate $Q$ value among various preferences. To update $Q$ for $i$th agent, we fix $\pi$, critic network is updated by the objective:

$$\max_Q \mathcal{J}(Q,i) = \mathbb{E}_{\rho_{E_i}} \left[ \phi \big( \omega_i^T \cdot (Q(s,a,\omega_i) - \gamma \mathbb{E}_{s' \sim \mathcal{P}(\cdot|s,a)} V^{\pi_i}(s',\omega_i)) \big) \right]$$
$$- (1-\gamma)\mathbb{E}_{p_0}[\omega_i^T \cdot V^{\pi_i}(s_0)] - \beta \sum_{j=i-1}^{i} ||\boldsymbol{r}_j - \boldsymbol{r}_{j+1}||_2 \tag{9}$$

where $\boldsymbol{r_i} = \mathcal{T}^\pi Q_i$ is the estimated vector reward of $i$th agent.

**Actor network update:** We use $\pi(s,a,\omega_i) \approx \pi_i(s,a)$. For a fixed $Q$ and $\omega_i$, we update $\pi$ for $i$th agent by minimizing the expected KL-divergence (Haarnoja et al., 2018):

$$\min_\pi \mathcal{J}(\pi,i) = \mathbb{E}_{s \sim \mathcal{D}_i, a \sim \pi(\cdot|s,\omega_i)} \big[ \log \pi(a|s,\omega_i) - \omega_i^T \cdot Q(s,a,\omega_i) \big] \tag{10}$$

where $\mathcal{D}_i$ is the distribution of previously sampled states or a replay buffer of $i$th expert and agent.

---

**Algorithm 1** Multi-Objective Inverse soft-Q Learning (MOIQ)

---

Initialize networks $Q_\phi$ and $\pi_\psi$
**while** environment step $t \leq$ N **do**
    **for** each expert $i$ **do**
        **for** each episode step in [1, T] **do**
            $a_t \sim \pi(\cdot|s_t, \omega_i)$
            $s_{t+1} \sim \mathcal{P}(\cdot|s_t, a_t)$
            $\mathcal{D}_i \leftarrow \mathcal{D}_i \cup \{(s_t, a_t, s_{t+1})\}$
            Update $Q_\phi$ according to equation 9
            $\phi_{t+1} \leftarrow \phi_t + \lambda_Q \nabla_\phi \mathcal{J}(Q,i)$
            Update $\pi_\psi$ according to equation 10
            $\psi_{t+1} \leftarrow \psi_t - \lambda_\pi \nabla_\psi \mathcal{J}(\pi,i)$
        **end for**
        $t \leftarrow t + T$
    **end for**
**end while**

---

## 5 EXPERIMENTS

### 5.1 EXPERTS

For discrete DST, an optimal stochastic policy is adopted to collect demonstrations. Specifically, let $d_x^b, d_y^b$ be the distances to the border of the current grid along x and y axis, $d_x^t, d_y^t$ be the distances to the target treasure of the current grid along x and y axis. The probability of going right or down is proportional to the $\min(d_x^b, d_x^t)$ and $\min(d_y^b, d_y^t)$ of the current grid. For continuous DST and Mujoco environments, the experts are trained from scratch with SAC for each distinct preference for 0.5M steps.

**Experts' preferences:** We prepare these experts with various preferences for each environment.

- Discrete DST MiniMap: [0.9, 0.1], [0.1, 0.9]

- Discrete DST DefaultMap: [0.9, 0.1], [0.5, 0.5], [0.1, 0.9]

- Continuous DST: [0.9, 0.1], [0.5, 0.5], [0.1, 0.9]

- Mo-Hopper: [0.8, 0.1, 0.1], [0.1, 0.8, 0.1], [0.1, 0.1, 0.8]

- Mo-Walker: [0.9, 0.1], [0.5, 0.5], [0.1, 0.9]

- Mo-HalfCheetah: [0.9, 0.1], [0.5, 0.5], [0.1, 0.9]

- Mo-Ant: [0.9, 0.1], [0.5, 0.5], [0.1, 0.9]

## 5.2 ENVIRONMENTS

For discrete DST, Mo-HalfCheetah, Mo-Hopper, we directly use Alegre et al. (2022), which is a multi-objective gymnasium environment. For continuous DST, we modify both state and action space of discrete DST to 2-dimensional continuous space, indicating position and velocity respectively. For Mo-Walker and Mo-Ant, we inherit the classes of Walker2d and Ant from Towers et al. (2023) and extend the reward space to two dimension. Information of each dimension of reward and further details are listed below.

**DST:** 2-dimensional reward space in the form *(treasure value, step cost)*, where treasure value is designed by Yang et al. (2019) and step cost is $-1$ for each step.

**Mo-Hopper:** 3-dimensional reward space in the form *(velocity in x-axis, height, control cost)* with the healthy reward $+1$ is directly added to every dimension of reward if the agent is healthy at timestep $t$.

**Mo-Walker:** 2-dimensional reward space in the form *(velocity in x-axis, control cost)* with the healthy reward $+1$ is directly added to every dimension of reward if the agent is healthy at timestep $t$.

**Mo-HalfCheetah:** 2-dimensional reward space in the form *(velocity in x-axis, control cost)*.

**Mo-Ant:** 2-dimensional reward space in the form *(velocity in x-axis, control cost)* with the healthy reward $+1$ is directly added to every dimension of reward if the agent is healthy at timestep $t$.

## 5.3 RESULTS

Since there are few IRL algorithms with multi-expert setting, we compare our results with GAIL (Ho & Ermon, 2016). In GAIL, we separately train 3 models for different preferences in one environment with 10 expert demos from each preference. In MOIQ, we train a single model with constraint coefficient $\beta = 5$ for different preferences in an environment with 10 expert demos from each preference, 30 expert demos is used totally.

As shown in Figure 2, MOIQ is competitive with GAIL in all 5 environments. Besides, In contrast to GAIL, MOIQ enjoys a faster learning rate and more sample-efficient. Take DST environment for instance, GAIL isn't competitive here. It's probably because the lack of demonstrations. Unlike experts in Mujoco environment where an near-optimal policy would have an average steps around one thousand in one episode, the expert at *DST - [0.1, 0.9]* take only 2 steps to the terminal state, resulting in 20 state-action pairs for 10 expert demos. However, MOIQ can reach expert-level performance within 100K environment steps with the same amount of demonstrations given to each preference.

**Expert-like performance:** We save our model as checkpoint every 5000 environment steps and pick the best model in terms of average return of the evaluation. As demonstrated in table 1, our model almost achieves expert-level performance in every preference of different environment. 4 out of 15 settings can even beat the experts.

## 5.4 TRANSFERABILITY

As shown in Figure 3, we demonstrate the transferability of our model by visualizing return in two dimensions for environments with 2-dimensional reward space. In DST and Mo-Ant, even with only demonstrations in 3 different preference, our model can still act correctly according to the preference given. It doesn't solely rely on the powerful approximation capability of neural networks but also significantly contributes to the precision of the learned reward.

In Mo-Walker and Mo-HalfCheetah, although it also achieves a descent scalar return, the visualization results show that the preference doesn't match the vectorized return quite well. This misalignment likely comes from the fact that the trained experts do not exhibit sufficient distinction in terms of the two-dimensional return according to their preferences.

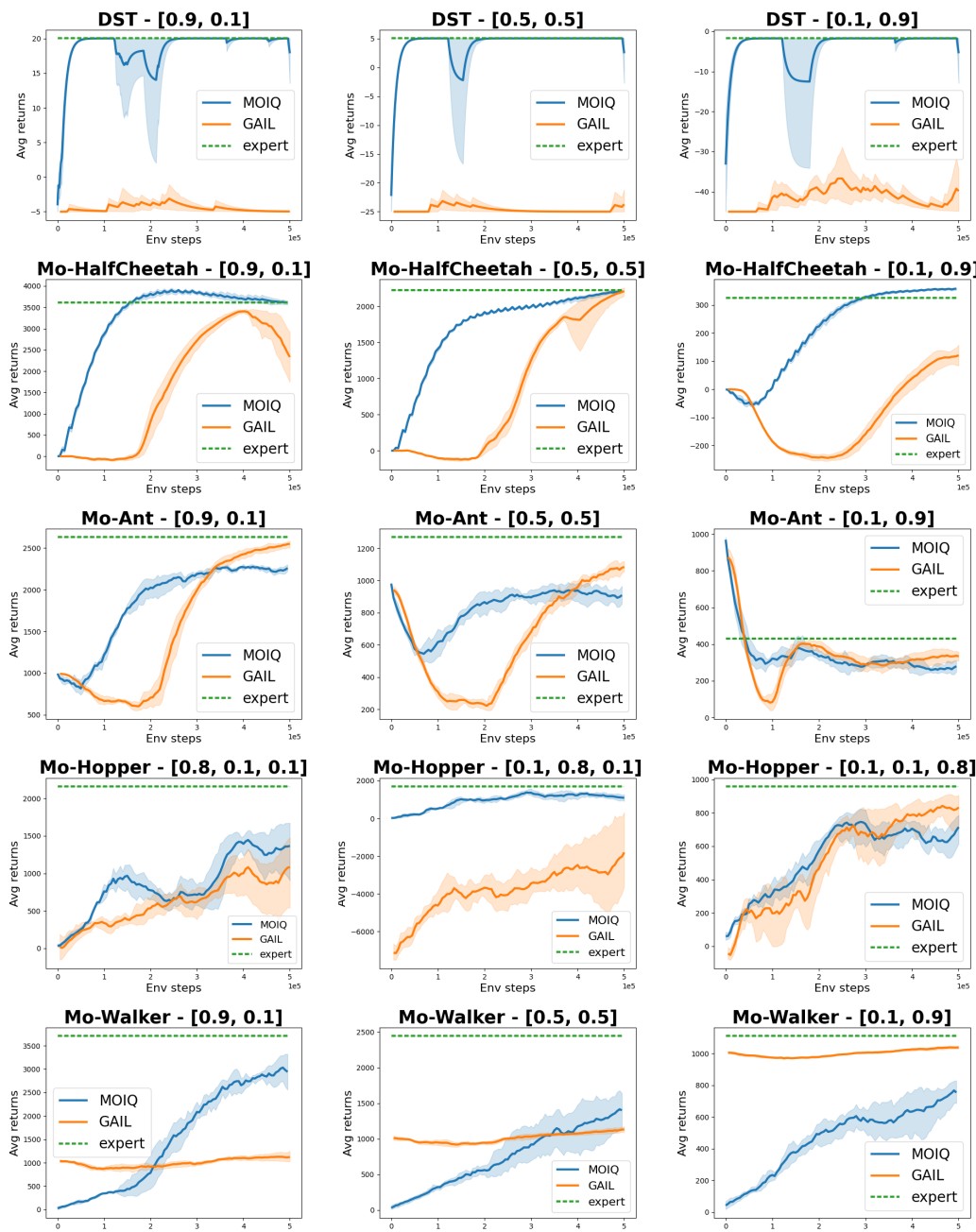

Figure 2: **Evaluation results while training.** Results are averaged from 5 different seeds and smoothed by taking ewma return with *alpha*=0.1

# 6 DISCUSSION

**Limitations:** The major limitation of our model lies in the quality of demonstrations. While these demonstrations need not be optimal, they must show sufficient distinctiveness in order to illustrate their differences in certain dimensions of the reward from others. Another limitation lies in its reliance on experts' preferences, making it a bit harder to collect datasets with labeled preferences.

**Future work:** One of our top priority must be learning preferences of experts, allowing our method to truly move away from hand-crafted components, including rewards and preferences. We find

| Env | Preference | MOIQ (Ours) | Expert |
|---|---|---|---|
| Continuous DST | [0.9, 0.1] | $20.03 \pm 0$ | $20.03 \pm 0$ |
| | [0.5, 0.5] | $5.05 \pm 0$ | $5.05 \pm 0$ |
| | [0.1, 0.9] | $-1.73 \pm 0$ | $-1.73 \pm 0$ |
| Mo-HalfCheetah | [0.9, 0.1] | $\mathbf{4377 \pm 41}$ | $3611 \pm 75$ |
| | [0.5, 0.5] | $\mathbf{2261 \pm 36}$ | $2223 \pm 18$ |
| | [0.1, 0.9] | $315 \pm 15$ | $\mathbf{325 \pm 8}$ |
| Mo-Hopper | [0.8, 0.1, 0.1] | $2055 \pm 212$ | $\mathbf{2155 \pm 99}$ |
| | [0.1, 0.8, 0.1] | $\mathbf{2283 \pm 201}$ | $1686 \pm 157$ |
| | [0.1, 0.1, 0.8] | $896 \pm 9$ | $\mathbf{958 \pm 8}$ |
| Mo-Walker | [0.9, 0.1] | $3577 \pm 225$ | $\mathbf{3706 \pm 64}$ |
| | [0.5, 0.5] | $1735 \pm 353$ | $\mathbf{2442 \pm 55}$ |
| | [0.1, 0.9] | $879 \pm 133$ | $\mathbf{1110 \pm 36}$ |
| Mo-Ant | [0.9, 0.1] | $2475 \pm 68$ | $\mathbf{2629 \pm 26}$ |
| | [0.5, 0.5] | $1039 \pm 134$ | $\mathbf{1269 \pm 12}$ |
| | [0.1, 0.9] | $\mathbf{463 \pm 117}$ | $431 \pm 12$ |

Table 1: **Testing return of the best-performance model.** Evaluations of return of MOIQ are conducted over 100 episodes, and the results are averaged across 5 different seeds. Experts' result are averaged from 10 demonstrations given.

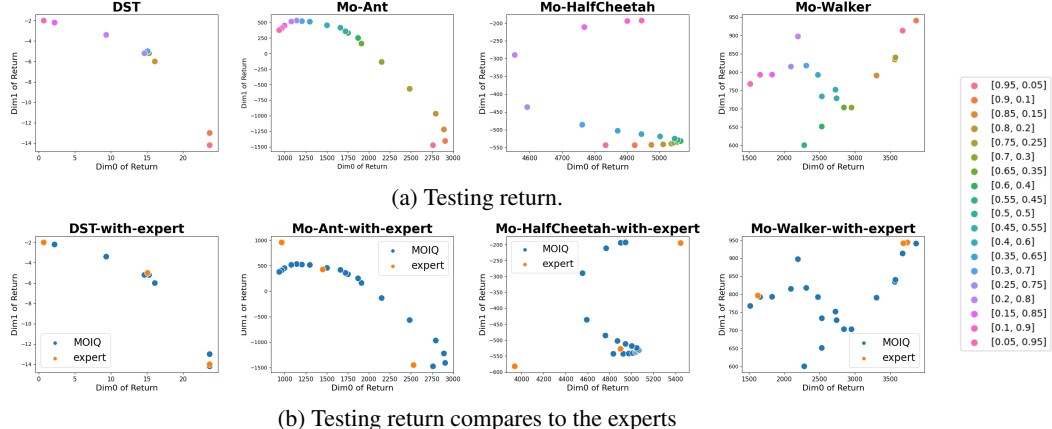

(a) Testing return.

(b) Testing return compares to the experts

Figure 3: **Transferability of the best-performance model.** Each point is obtained by feeding in a specific preference value from $[1 - 0.05 \times i, 0.05 \times i]$ for $i \in [1, 19]$. Evaluations are conducted over 100 episodes, and the results are averaged across 5 different seeds.

this task particularly challenging because it's not an easy optimization problem. Preference is a relative concept that requires comparing with others, which might have profound connections with this work. We're looking forward to working on this topic in the future.

## 7 CONCLUSION

We have seen the needs of considering multiple heterogeneous experts in IRL. Enlightened by this, we assume common reward is the bridge that connects every agent together. We first conduct a simple and meaningful experiment on discrete environment in order to demonstrate that the idea of common reward does work. We then propose MOIQ $-$ an approach integrating the common reward constraint into the critic objective. By turning the weakness of heterogeneous demonstrations into strength, it can compete with GAIL in terms of sample efficiency and average return in continuous DST and Mujoco environment.

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

# A APPENDIX

## A.1 ADDITIONAL RESULTS

**Ablation on constraint coefficient:** As shown in table 2, we present our ablation studies on constraint coefficient by comparing the results of $\beta = 0$ and $\beta = 5$. 10 out of 15 settings show improvements with $\beta = 5$.

| Env | Preference | $\beta = 0$ | $\beta = 5$ | Percentage change |
|---|---|---|---|---|
| Continuous DST | [0.9, 0.1] | 19.96 | 19.927 | -0.17% |
| | [0.5, 0.5] | 5.044 | 5.044 | 0.00% |
| | [0.1, 0.9] | -1.736 | -1.736 | 0.00% |
| Mo-HalfCheetah | [0.9, 0.1] | 3578 | 3654 | 2.12% |
| | [0.5, 0.5] | 2152 | 2160 | 0.37% |
| | [0.1, 0.9] | 341 | 347 | 1.76% |
| Mo-Hopper | [0.8, 0.1, 0.1] | 1118 | 1268 | 13.42% |
| | [0.1, 0.8, 0.1] | 1081 | 1189 | 9.99% |
| | [0.1, 0.1, 0.8] | 638 | 707 | 10.82% |
| Mo-Walker | [0.9, 0.1] | 2386 | 2636 | 10.48% |
| | [0.5, 0.5] | 1560 | 1326 | -15.00% |
| | [0.1, 0.9] | 714 | 738 | 3.36% |
| Mo-Ant | [0.9, 0.1] | 2265 | 2249 | -0.71% |
| | [0.5, 0.5] | 926 | 911 | -1.62% |
| | [0.1, 0.9] | 295 | 281 | -4.75% |

Table 2: **Comparison between $\beta = 0$ and $\beta = 5$.** The numerical results are ewma returns with *alpha*=0.1 and averaged across 5 different seeds. In DST, we show the results at 100K environment steps. In Multi-objective Mujoco environments, we show the results at 500K environment steps.

**Reward correlations:** We show the Pearson correlation between the true reward and the learned reward in table 3.

| Env | Preference | Correlation |
|---|---|---|
| Mo-HalfCheetah | [0.9, 0.1] | 0.63 |
| | [0.5, 0.5] | 0.52 |
| | [0.1, 0.9] | 0.19 |
| Mo-Hopper | [0.8, 0.1, 0.1] | 0.91 |
| | [0.1, 0.8, 0.1] | 0.1 |
| | [0.1, 0.1, 0.8] | 0.93 |
| Mo-Walker | [0.9, 0.1] | 0.96 |
| | [0.5, 0.5] | 0.96 |
| | [0.1, 0.9] | 0.98 |
| Mo-Ant | [0.9, 0.1] | 0.92 |
| | [0.5, 0.5] | 0.95 |
| | [0.1, 0.9] | 0.86 |

Table 3: **Pearson correlation coefficient:** It measures linear correlation between two sets of data.

**Does MOIQ really improve?** We show the learning curve comparing to IQ-Learn(Garg et al., 2022). In IQ-Learn, we separately train 3 models for different preferences in one environment with 1 expert demos from each preference. In MOIQ, we train a single model with constraint coefficient $\beta = 5$ for different preferences in an environment with 1 expert demos from each preference, 3 expert demos is used totally.

As shown in Figure 4. In Mo-Ant, IQ-Learn suffers from a serious instability issue while MOIQ is growing steadily. It clearly shows the benefits from connecting multiple expert with common reward functions. Another thing to highlight is that IQ-Learn or other traditional IRL methods takes

$n$ times more environment steps than MOIQ in an environment, where $n$ is the number of experts. Despite this, MOIQ can always achieve almost the same or superior performance compared to IQ-Learn. It can be contributed to the fact that they are trained with a single network in MOIQ. A single model works because these experts are involved in the same task, with the only differences being at preference level. There's no need to train $n$ independent models for each expert.

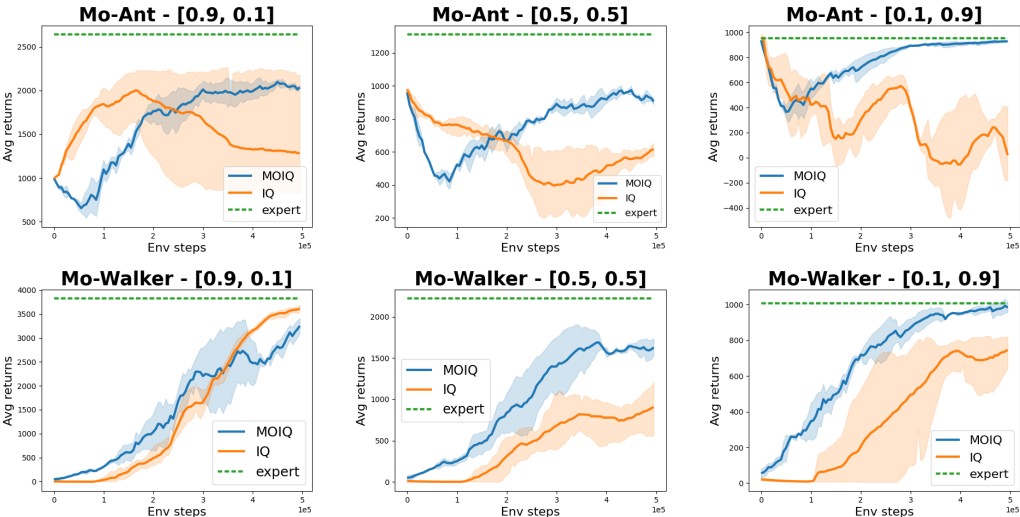

Figure 4: **Evaluation results while training.** Results are averaged from 3 different seeds and smoothed by taking ewma return with *alpha*=0.1. Note that the experts used here are not exactly same as in Figure 2. Specifically, experts of Mo-Walker are trained from scratch for 1.5M steps using the SAC algorithm implemented in Spinning Up (Achiam, 2018).

## A.2 IMPLEMENTATION DETAILS

**GAIL:** We implement GAIL using imitation (Gleave et al., 2022), with SAC as the generator.

| Parameter | Value |
|---|---|
| Policy | MlpPolicy |
| Learning rate | 3e-4 |
| Buffer size | 1e6 |
| Batch size | 256 |
| Tau | 0.05 |
| Gamma | 0.99 |
| Train frequency | 1 |

Table 4: **Hyperparameters of GAIL**

**MOIQ and Experts:** We implement our algorithm based on the open-source code of IQ-Learn (Garg et al., 2022). Its implementation is built on SAC, the hyperparameters used are listed in table 5.

| Parameter | Value |
|---|---|
| Policy | MlpPolicy |
| Hidden dim | [255, 255] |
| Critic lr | 3e-4 |
| Actor lr | 3e-5 |
| Buffer size | 1e6 |
| Batch size | 256 |
| Critic update frequency | 1 |
| Actor update frequency | 1 |
| Critic tau | 0.005 |

Table 5: **Hyperparameters of MOIQ**

