# OpenReview forum: "Learning From Multi-Expert Demonstrations: A Multi-Objective Inverse Reinforcement Learning Approach"
_ICLR.cc/2024/Conference — Submitted to ICLR 2024_

### Official Review · Reviewer_92m6 · 2023-10-30

**Soundness:** 2 fair
**Presentation:** 2 fair
**Contribution:** 2 fair
**Rating:** 3
**Confidence:** 3

**Summary:**

Summary: The authors propose a multi-objective inverse reinforcement learning approach that can learn from multiple experts. Towards this purpose, the authors formulate an approach that learns the common reward across demonstrators utilizing the alternating direction method of multipliers (ADMM) in the discrete case and build on IQ-Learn in the continuous case. The authors present details regarding their approach and preliminary work. Finally, the authors conclude by presenting an evaluation in several domains by creating varying-preferenced demonstrators as the training data. The results are positive in some domains, with the quality of demonstrator imitation varying given the preference.

**Strengths:**

This paper has several strengths:
+ To the best of my knowledge, solving for the common reward with consensus ADMM is novel. Further extending the IQ-learn framework to the field of MOIRL is novel.
+ Presenting results on each preference individually is beneficial, paints a picture of the algorithm's full performance, and can display features such as mode collapse.

**Weaknesses:**

- The novelty of this paper is not clear with respect to a string of prior works that utilize a very similar approach (inferring a common reward function to learn from heterogeneous demonstrations). Could you clarify how your work differs from the work below?
1. Joint Goal and Strategy Inference across Heterogeneous Demonstrators via Reward Network Distillation (https://arxiv.org/abs/2001.00503)
2. Fast Lifelong Adaptive Inverse Reinforcement Learning from Demonstrations (https://arxiv.org/abs/2209.11908)
3. InfoGAIL
- Further, as learning from heterogeneous demonstrators has been studied for several years, it would be beneficial to compare against a framework that attempts to accomplish this goal. It would be expected that a baseline like GAIL will underperform the proposed approach.
- It isn't clear how the preferences were designed in Section 5 and how the preferences result in different behaviors qualitatively.
- Justifications should be added for the critic and policy update simplifications in 4.2.2.
- The results would benefit from more detail and justification. It is unclear why the proposed approach underperforms GAIL in some instances.
- The paper could benefit from a read-through to improve clarity and grammar.

**Questions:**

Could you please address the weaknesses mentioned above?

---

> ### Author Response · Authors · 2023-11-20
>
> We appreciate the constructive feedbacks from you. Here, we’d like to address the following points:
>
> * **Could you clarify how your work differs from the work below?**
>
> 1. Joint Goal and Strategy Inference across Heterogeneous Demonstrators via Reward Network Distillation [1]
>
>     The core idea of MSRD is connecting $n$ separate IRL through a regularization term, thereby recovering a robust task reward. The biggest problem is it completely ignores the computational cost because it still needs to run IRL $n$ times, which produces $n+1$ independent reward networks and $n$ independent policy networks. Not to mention, at the end of each iteration, it still need to run TRPO to determine $\pi_i$.
>
> 2. Fast Lifelong Adaptive Inverse Reinforcement Learning from Demonstrations [2]
>
>     Although FLAIR slightly reduces the computation loss through policy mixture, it still needs to run IRL multiple times when a new expert is unique enough. In contrast, Our work adopts a single model architecture, as stated in the abstract. The core idea of our work is sharing knowledge within a single model. The reason behind this is that these heterogeneous experts are engaging in the same task, with their only differences lying at the preference level.
>
> 3. InfoGAIL [3]
>
>     Although InfoGAIL and our work share a similar motivation in addressing the heterogeneity of experts, InfoGAIL introduces a latent variable that can only be used to disentangle trajectories. In contrast, our work can recover a common reward that has transferability to different preferences. Besides, InfoGAIL is constrained by the limitations of IL, such as the inability to adapt to environmental changes and excessive reliance on the quantity and quality of experts.
>
>
> * **Further, as learning from heterogeneous demonstrators has been studied for several years, it would be beneficial to compare against a framework that attempts to accomplish this goal.**
>
> In the updated version, we have included IQ-Learn [4], the base model of our work, as another baseline to further demonstrate the improvements. Please refer to appendix A.1 "Does MOIQ really improves?"
>
>
>
> * **It isn't clear how the preferences were designed in Section 5 and how the preferences result in different behaviors qualitatively.**
>
> As we focus on the setting of multi-expert (heterogeneous) in this work, it would have very limited use picking similar preferences. The preferences designed in 5.1 aim to accentuate heterogeneities among experts as much as possible while taking different dimensions of the reward function into account.
>
>
> Thanks again for your efforts on reviewing our work. Hope that we have addressed all the raised issues. Let us know if you have any further problems.
>
> **References:**
> 1. Chen, Letian, et al. "Joint goal and strategy inference across heterogeneous demonstrators via reward network distillation." Proceedings of the 2020 ACM/IEEE international conference on human-robot interaction. 2020.
> 2. Chen, Letian, et al. "Fast lifelong adaptive inverse reinforcement learning from demonstrations." Conference on Robot Learning. PMLR, 2023.
> 3. Li, Yunzhu, Jiaming Song, and Stefano Ermon. "Infogail: Interpretable imitation learning from visual demonstrations." Advances in neural information processing systems 30 (2017).
> 4. Garg, Divyansh, et al. "Iq-learn: Inverse soft-q learning for imitation." Advances in Neural Information Processing Systems 34 (2021): 4028-4039.

---

> > ### Comment · Reviewer_92m6 · 2023-11-21
> > **Rebuttal Response**
> >
> > Thank you for your response and the inclusion of an additional baseline. While the comparison against IQ-learn is helpful, a comparison to InfoGAIL would have been preferred as InfoGAIL attempts to learn from multi-preference expert data.

---

### Official Review · Reviewer_8hYp · 2023-11-03

**Soundness:** 3 good
**Presentation:** 2 fair
**Contribution:** 2 fair
**Rating:** 5
**Confidence:** 4

**Summary:**

This paper proposes an inverse reinforcement learning (IRL) framework when the reward function is a vector quantity and the expert demonstration consists of multiple experts. Using the key assumption that those experts share the common reward function, the authors extend the standard IRL framework into a multi-objective case with a linear scalarization function. This MOIRL adopts either the ADMM method for consensus or the inverse deep soft-q learning technique. Numerical results show that the proposed MOIQ algorithm produces a single model covering various expert demos from different preferences.

**Strengths:**

- This paper addresses multi-objective inverse reinforcement learning (MOIRL) framework, which seems to provide a new perspective.
- The paper provides some mathematical formula regarding the framework.
- Based on ADMM and inverse soft-Q learning, the proposed methodology gives technical soundness.

**Weaknesses:**

There are several fundamental discussions regarding the assumption of the proposed method, most of which is not clarified to me.
1) Is it valid in practice if we assume we know the number of multiple experts which produced the whole demonstration?
2) Is it valid in practice if we assume each multiple experts produced equal contribution to produce the whole given demonstration (since the formulation?
3) Is it valid in practice if we assume we know each preference (omega_i) of each expert? It seems that using 3 expert of [0.1,0.9], [0.5,0.5],[0.9,0.1] is a strong assumption.

4. In related work, two methods of MOIRL are introduced. Should these be encompassed in the baseline algorithm?

5. Can the proposed method applicable to other algorithm than SAC framework?

**Questions:**

Please check the above weakness part. Additional questions are as follows.

6. Can we use MOIQ for discrete case?

7. In Table 1, how MOIQ sometimes outperform expert?

---

> ### Author Response · Authors · 2023-11-20
>
> We appreciate the constructive feedbacks from you. Here, we’d like to address the following points:
>
> * **Is it valid in practice if we assume we know the number of multiple experts which produced the whole demonstration?**
>
> This assumption is relatively fair as long as we know the indices of experts. It's also used in ILEED [1]. If the assumption is removed, meaning we don't know both the number and preferences of experts (as preferences can be treated as an identifier), then two possible options may be considered:
> 1. Treat every trajectories heterogeneous initially.
> 2. Apply a preliminary clustering algorithm to trajectories.
>
>
> * **Is it valid in practice if we assume each multiple experts produced equal contribution to produce the whole given demonstration (since the formulation?**
>
> In our work, we assume that the experts are equally optimal. If that's not the case, it would require more meta-information from experts, such as the relative ranking or importance weight among experts, which may not be easy to acquire.
>
> * **Is it valid in practice if we assume we know each preference (omega_i) of each expert? It seems that using 3 expert of [0.1,0.9], [0.5,0.5],[0.9,0.1] is a strong assumption.**
>
> We agree that the assumption of preference-known could be a bottleneck of our work. Currently, we're working on enabling our model to learn preferences. One of the approaches we have tried is finding the one that maximizes the expected value of the estimated cumulated return during training, expressed as $\arg \max_\omega \mathbb{E}_{\tau_E}[ \sum _{t=0}^N \gamma^t \omega^T \cdot \hat{R}(s_t,a_t)]$. However, this approach didn't yield the desired results, as it would allocate the entire budget to the dimension with the higher estimated cumulated return, which seems to be unreasonable. We'd love to hear your opinions and suggestions to help us enhance our work.
>
>
> * **In related work, two methods of MOIRL are introduced. Should these be encompassed in the baseline algorithm?**
>
> For MODIRL [3], we didn't found the realeased official code. For ILEED, they did release the official code for discrete action space. We have reached out to them to ask for the code applicable for continuous action space, and the reply we received was *there is no working implementation that we can easily pull*. However, they did provide us with detailed instructions to implement it ourselves, and there are still some obstacles to overcome. As an alternative option, we have included IQ-Learn [5], the base model of our work, as another baseline to further demonstrate the improvements. Please refer to appendix A.1 "Does MOIQ really improves?"
>
>
> * **Can the proposed method applicable to other algorithm than SAC framework?**
>
> Indeed, the concept of the *common reward* can be done in various approaches. The key is how you enforce the constraint of common reward. In section 4.1, we demonstrate that the common reward can be found by using consensus ADMM. Subsequently, we run PPO [4] to find the corresponding policy.
>
>
> * **Can we use MOIQ for discrete case?**
>
> Sure. It works in discrete case with some appropriate modifications on SAC [2]. E.g., The output of actor in SAC is originally a Gaussian distribution. It should be modified to output a probability of each action.
>
>
> * **In Table 1, how MOIQ sometimes outperform expert?**
>
> It's one of the benefits when considering multiple experts. Having a highly correlated estimated reward function allows MOIQ to find the best model that connects experts. It has the potential to outperform the original expert. This is something that cannot be accomplished with single-expert.
>
> Thanks again for your efforts on reviewing our work. Hope that we have addressed all the raised issues. Let us know if you have any further problems.
>
> **References:**
>
> 1. Beliaev, Mark, et al. "Imitation learning by estimating expertise of demonstrators." International Conference on Machine Learning. PMLR, 2022.
> 2. Haarnoja, Tuomas, et al. "Soft actor-critic: Off-policy maximum entropy deep reinforcement learning with a stochastic actor." International conference on machine learning. PMLR, 2018.
> 3. Kishikawa, Daiko, and Sachiyo Arai. "Multi-objective deep inverse reinforcement learning through direct weights and rewards estimation." 2022 61st Annual Conference of the Society of Instrument and Control Engineers (SICE). IEEE, 2022
> 4. Schulman, John, et al. "Proximal policy optimization algorithms." arXiv preprint arXiv:1707.06347 (2017).
> 5. Garg, Divyansh, et al. "Iq-learn: Inverse soft-q learning for imitation." Advances in Neural Information Processing Systems 34 (2021): 4028-4039.

---

> > ### Comment · Reviewer_8hYp · 2023-11-22
> > **Thanks**
> >
> > Thanks for the reply. Although the authors dealt with several issues I raised, I also agree with other reviewers concerns. Therefore, I maintain my original rating.

---

### Official Review · Reviewer_ooq3 · 2023-11-03

**Soundness:** 2 fair
**Presentation:** 2 fair
**Contribution:** 2 fair
**Rating:** 3
**Confidence:** 4

**Summary:**

This paper introduces a multi-objective Inverse Reinforcement Learning (IRL) approach within the framework of Alternating Direction Method of Multipliers (ADMM). This novel approach not only converges towards optimal solutions but also attains global solutions. The authors establish a link to Inverse Soft Q-Learning and illustrate the efficacy of their algorithm through experimentation on Mo-Mujoco and DST environments.

**Strengths:**

* Leveraging the theory of ADMM as a foundational framework for Multi-Objective Markov Decision Processes offers an intriguing perspective. This approach enables authors to train a single model, eliminating the need for multiple models catering to distinct preferences.

* The authors have developed an approach, grounded in the inverse Bellman operator, that can efficiently scale to both discrete and continuous tasks.

**Weaknesses:**

* The related works are quite limited. A more thorough review should be presented.

* The experimental results, while valuable, would benefit from a more comprehensive analysis. Currently, the authors only compare their proposed algorithms to Generative Adversarial Imitation Learning (GAIL), and it's worth noting that in the Mo-walker and Mo-Ant environments, the Average Return of MOIQ performs worse than GAIL. Additional analyses are necessary to provide a more complete evaluation of these experiments.

* The author's use of the variable 'w' is somewhat ambiguous as it is defined multiple times in the paper. It is unclear whether 'w' represents the preference of the expert or a d-dimensional probability vector. Clarification is needed on this aspect.

**Questions:**

* Could the authors provide a clear definition of the symbol $\rho$ in Equation (3), and elaborate on the method for determining its value?"

* The authors impose a constraint that $r_i = r$ even when expert demonstrations exhibit heterogeneity. Does the use of a common reward function imply the treatment of all demonstrations as homogeneous?

* It would be valuable if the authors could provide a more in-depth explanation regarding the instability observed in the transfer experiments, particularly in the Mo-HalfCheetah and Mo-Walker environments.

* Could the author please illustrate the distinction between the common reward function and the ground-truth reward function?

---

> ### Author Response · Authors · 2023-11-20
>
> We appreciate the constructive feedbacks from you. Here, we’d like to address the following points:
>
> * **The related works are quite limited. A more thorough review should be presented.**
>
> After carefully reading through the valuable feedbacks from reviewers, we have provided a more comprehensive review in our related works.
>
> * **The experimental results, while valuable, would benefit from a more comprehensive analysis.**
>
> In the updated version, we have included IQ-Learn [1], the base model of our work, as another baseline to further demonstrate the improvements. Please refer to appendix A.1 "Does MOIQ really improves?"
>
>
> * **The author's use of the variable $\omega$ is somewhat ambiguous as it is defined multiple times in the paper. It is unclear whether $\omega$ represents the preference of the expert or a d-dimensional probability vector. Clarification is needed on this aspect.**
>
> Thanks for pointing out this misleading definition, we have corrected it in our updated version. The confusion may come from the word "probability". We refer to $d$-dimensional probability vector as a vector with $d$ non-negative entries that adds up to $1$. Furthermore, we use it to denote the preferences of experts.
>
> * **Could the authors provide a clear definition of the symbol $\rho$ in Equation (3), and elaborate on the method for determining its value?**
>
> $\rho$ is the positive penalty parameter in ADMM, responsible for managing the trade-off between primal and dual consistency. The value of $\rho$ significantly influences the algorithm's convergence speed and stability. When $\rho$ is excessively large, it may lead to instability, while a too-small $\rho$ can result in a slower convergence of the algorithm.
>
> * **The authors impose a constraint that $r_i=r$ even when expert demonstrations exhibit heterogeneity. Does the use of a common reward function imply the treatment of all demonstrations as homogeneous?**
>
> Even though we impose a common reward constraint for the reward vector, the scalarized reward can be different given different preferences.
>
> * **It would be valuable if the authors could provide a more in-depth explanation regarding the instability observed in the transfer experiments, particularly in the Mo-HalfCheetah and Mo-Walker environments.**
>
> As described in section 5.1, the experts in Mujoco are trained from scratch for 0.5M steps with their own scalarized rewards, computed by eq (1). They are expected to acquire corresponding amounts in each dimension of return according to their preferences, like the orange points in DST and Mo-Ant in Fig 3(b). However, the trained experts didn't go the way we expected, resulting in a limited differences among experts. Therefore, it influenced the transferability in our model.
>
>
> * **Could the author please illustrate the distinction between the common reward function and the ground-truth reward function?**
>
> In this work, you can view them as same things. More specifically, we try to emphasize our differences with traditional IRL. In traditional IRL, we can still recover one reward function for an expert. If there are $n$ experts, then we can recover $n$ reward functions by repeating IRL $n$ times for every expert's demonstration. However, these $n$ reward functions can be greatly different. That's why we emphasize common reward function in our MOIRL work.
>
> Thanks again for your efforts on reviewing our work. Hope that we have addressed all the raised issues. Let us know if you have any further problems.
>
> **References:**
> 1. Garg, Divyansh, et al. "Iq-learn: Inverse soft-q learning for imitation." Advances in Neural Information Processing Systems 34 (2021): 4028-4039.

---

### Official Review · Reviewer_YYg8 · 2023-11-06

**Soundness:** 2 fair
**Presentation:** 2 fair
**Contribution:** 2 fair
**Rating:** 3
**Confidence:** 4

**Summary:**

This paper addresses the challenge of utilizing imitation learning in real-world scenarios, particularly when faced with multiple sources of expert data. The authors frame this issue as a multi-objective inverse reinforcement learning problem and introduce a method that exhibits transferability across various preferences. Their experimental findings demonstrate that this approach not only yields competitive results with baseline but also requires fewer computational resources.

**Strengths:**

1. The paper's notable strength lies in its dedicated attention to a pivotal problem, with the potential to enhance the effectiveness of imitation learning when applied in real-world scenarios.

**Weaknesses:**

1. Annotating the preference vector $\omega$ can be cumbersome, especially when dealing with a large expert demonstration dataset collected from multiple human expert sources. This manual annotation process may become a bottleneck, particularly when preferences vary significantly among the human experts.

2. The proposed method primarily addresses the challenge of demonstration diversity stemming from different preferences among experts, but it does not explicitly tackle the diversity arising from multi-modality or stochastic behavior within the same preference category. For instance, some experts might have the same preference but choose different paths, such as passing by a tree on the left or the right.

3. The choice of the baseline method appears to be relatively weak. While multi-expert inverse reinforcement learning algorithms are limited, the authors could have selected more advanced IRL methods as baselines and trained a model for each preference individually. This would have provided a better understanding of the proposed method's capabilities by allowing for a more robust comparison.

**Questions:**

1. What factors contribute to the notably superior performance of MOIQ and GAIL at the early stages of training, when compared to the expert's performance in the Mo-Ant environment, specifically in the preference setting [0.1, 0.9] as depicted in Figure 2?

---

> ### Author Response · Authors · 2023-11-20
>
> We appreciate the constructive feedbacks from you. Here, we’d like to address the following points:
>
> * **Annotating the preference vector omega can be cumbersome, especially when dealing with a large expert demonstration dataset collected from multiple human expert sources.**
>
> We agree that it could be a bottleneck of our work. Currently, we're working on enabling our model to learn preferences. One of the approaches we have tried is finding the one that maximizes the expected value of the estimated cumulated return during training, expressed as $\arg\max_\omega \mathbb{E}_{\tau_E}[\sum _{t=0}^N\gamma^t\omega^T \cdot \hat{R}(s_t,a_t)]$. However, this approach didn't yield the desired results, as it would allocate the entire budget to the dimension with the higher estimated cumulated return, which seems to be unreasonable. We'd love to hear your opinions and suggestions to help us enhance our work.
>
>
> * **It does not explicitly tackle the diversity arising from multi-modality or stochastic behavior within the same preference category.**
>
> The conducted experiments indeed involve experts with multi-modality. As stated in section 5.1, we adopt the optimal stochastic policy as experts in discrete DST environment. The results are shown in Figure 1.
>
>
> * **The choice of the baseline method appears to be relatively weak.**
>
> In the updated version, we have included IQ-Learn [1], the base model of our work, as another baseline to further demonstrate the improvements. Please refer to appendix A.1 "Does MOIQ really improves?"
>
> * **What factors contribute to the notably superior performance of MOIQ and GAIL at the early stages of training, when compared to the expert's performance in the Mo-Ant environment, specifically in the preference setting $[0.1, 0.9]$ as depicted in Figure 2?**
>
> There's a healthy reward $+1$ at each timestep $t$ in Mo-Ant. At the early phases of training, agents hardly move and therefore are rewarded $\sim 1000$ for surviving $1000$ steps (max timesteps). We have re-trained our expert with the preference setting $[0.1, 0.9]$ in Mo-Ant. Please refer to Figure 4 in appendix A.1.
>
> Thanks again for your efforts on reviewing our work. Hope that we have addressed all the raised issues. Let us know if you have any further problems.
>
> **References:**
> 1. Garg, Divyansh, et al. "Iq-learn: Inverse soft-q learning for imitation." Advances in Neural Information Processing Systems 34 (2021): 4028-4039.

---

### Meta-Review · Area_Chair_cjXn · 2023-12-07

**Metareview:**

The paper addresses challenges in imitation learning from multiple expert demonstrations. The authors propose a multi-objective inverse reinforcement learning (MOIRL) approach, incorporating the Alternating Direction Method of Multipliers (ADMM) and Inverse Soft Q-Learning frameworks. The common theme is the assumption of a shared reward among experts, enabling transferability across various preferences. The proposed methods demonstrate competitive results with baseline approaches while requiring fewer computational resources. Additionally, the MOIRL framework extends to cases where the reward function is a vector, achieving a single model capable of accommodating diverse expert demonstrations.

The paper addresses a key problem in imitation learning with practical applications. Its innovative use of ADMM for Multi-Objective Markov Decision Processes and the efficient scaling to both discrete and continuous tasks demonstrate versatility. Introducing a novel multi-objective inverse reinforcement learning (MOIRL) framework adds a fresh perspective, complemented by clear mathematical formulations.

On the other hand, the paper faces criticism for potential bottlenecks in manual annotation and overlooking multi-modality within the same preference. Critiques include the baseline method choice lacking advanced Inverse Reinforcement Learning (IRL) methods, a limited review of related works, and incomplete experimental analysis. The paper's novelty compared to similar works is questioned, and the absence of comparisons with frameworks like GAIL is noted. Overall, a read-through is recommended for enhanced clarity and grammar.

Even if the authors' rebuttals solved some of the issues raised in the reviews, the reviewers still consider this paper not ready for acceptance.

**Justification For Why Not Higher Score:**

The reviewers have identified several weaknesses, and they agree that the paper is not ready for publication.

**Justification For Why Not Lower Score:**

N/A

---

### Decision · Program_Chairs · 2024-01-16

Reject